# Anytime, Anywhere, Anyone: Investigating the Feasibility of Segment Anything Model for Crowd-Sourcing Medical Image Annotations

**Pranav Kulkarni**\*                                    PKULKARNI@SOM.UMARYLAND.EDU
**Adway Kanhere**\*                                      AKANHERE@SOM.UMARYLAND.EDU
**Dharmam Savani**\*                                     DSAVANI@SOM.UMARYLAND.EDU
**Andrew Chan**                                    ANDREW.CHAN@SOM.UMARYLAND.EDU
**Devina Chatterjee**                      DEVINACHATTERJEE@SOM.UMARYLAND.EDU
**Paul H. Yi**                                               PYI@SOM.UMARYLAND.EDU
**Vishwa S. Parekh**                                   VPAREKH@SOM.UMARYLAND.EDU
*University of Maryland Medical Intelligent Imaging (UM2ii) Center, Baltimore, MD 21201*

**Editors:** Accepted for publication at MIDL 2024

## Abstract

Curating annotations for medical image segmentation is a labor-intensive task that requires domain expertise, resulting in "narrowly" focused deep learning (DL) models with limited translational utility. We explore the potential of the Segment Anything Model (SAM) for crowd-sourcing "sparse" annotations from non-experts to generate "dense" segmentation masks for training 3D nnU-Net models. Our results indicate that while SAM-generated annotations exhibit high mean Dice scores compared to ground-truth annotations, SAM nnU-Net models perform significantly worse than ground-truth nnU-Net models.
**Keywords:** Data Annotation, Segment Anything Model, Foundation Model

## 1. Introduction

Developing deep learning (DL) models for medical image segmentation is an extremely labor-intensive task that requires a radiologist to manually annotate objects across a dataset (Diaz-Pinto et al., 2022; Wasserthal et al., 2023). As a result, most models developed in literature are "narrowly" focused on the task at hand with limited translational utility. Many approaches have been proposed where users can use less time-consuming "sparse" annotations (e.g., scribbles, bounding boxes) to interactively prompt a DL model to create "dense" mask annotations (Diaz-Pinto et al., 2022; Huang et al., 2018). While these approaches reduce annotation burden, an expert has to fine-tune and validate them. Therefore, there is a critical need for a data annotation pipeline that would allow non-experts to annotate datasets with sparse annotations without the need for an expert in the loop.

Recently, the Segment Anything Model (SAM) has revolutionized semantic segmentation with strong zero-shot generalizability across various domains, including medical imaging (Kirillov et al., 2023). It works by interactively prompting images with sparse annotations to generate segmentation masks. While recent literature suggests that SAM holds a lot of promise for annotating medical imaging datasets (Cheng et al., 2023; Mazurowski et al., 2023; Ma et al., 2024), it has yet to be evaluated in a realistic scenario. Our purpose is to 1) evaluate SAM for crowd-sourcing medical image annotations from non-experts, and 2) evaluate SAM-generated annotations for training 3D segmentation models.

---

\* Authors contributed equally to this work.

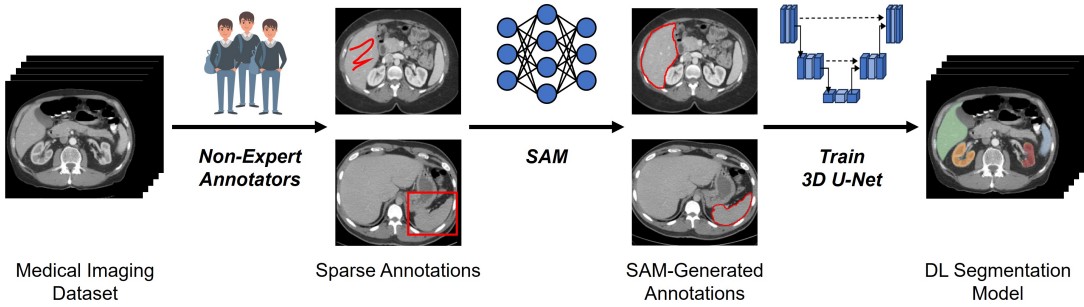

Figure 1: Pipeline for crowd-sourcing annotations for training 3D segmentation models.

## 2. Methods

**Dataset:** The Beyond the Cranial Vault (BTCV) dataset consists of $n = 30$ contrast-enhanced abdominal CT scans with annotations for 13 abdominal organs (Landman et al., 2015). We included five organs of interest: aorta, left and right kidneys, liver, and spleen. We randomly split the BTCV dataset into train and test sets ($n = 15$, both).

**Crowd-Sourcing Sparse Annotations:** We used the OpenLabeling tool (Cartucho et al., 2018) to annotate the BTCV train set. Each slice of a volume was annotated by four non-experts with small-to-moderate medical knowledge. They were provided with basic orientation and were instructed to draw bounding boxes surrounding the organs of interest. Each annotator was assigned one task and volumes were annotated independently.

**Generating Segmentation Masks:** We used the SAM ViT-Huge to generate segmentation masks for organs. Since SAM only supports 2D, each slice was passed as input with its corresponding boxes to SAM. The mask with the highest confidence score was selected. The SAM-generated annotations were then converted to NIfTI. We measured the mean Dice score of SAM-generated annotations on the ground-truth annotations of the train set.

**Training 3D Segmentation Models:** We train nnU-Net models, a self-configuring state-of-the-art (SOTA) 3D U-Net architecture (Isensee et al., 2021), on the SAM-generated (SAM-nnU-Net) and ground-truth annotations (GT-nnU-Net). The models were trained with 5-fold cross-validation for 1000 epochs using nnU-Net's training process with mirroring removed (Wasserthal et al., 2023). We compared their mean volume Dice scores on the ground-truth test set using Wilcoxon signed-rank tests. Statistical significance was defined as $p < 0.05$. Our code is available at: https://github.com/UM2ii/SAM_DataAnnotation

## 3. Results

The non-experts annotated 651 slices from the BTCV train set with 1840 bounding boxes (Figure 2). They took $55.60 \pm 8.76$ mins (mean $3.29 \pm 1.04$ secs per slice) to annotate an organ. However, only $n = 11$ volumes were fully annotated, with kidneys missing for $n = 3$ volumes, liver missing for $n = 2$ volumes, and spleen missing for $n = 1$ volume. We excluded the volumes with missing annotations. While SAM-generated annotations have a high slice Dice score (mean $0.88 \pm 0.02$), they have a lower volume Dice score (mean $0.70 \pm 0.09$). Furthermore, the SAM-nnU-Net model performs significantly worse than the GT-nnU-Net model ($0.80 \pm 0.05$ vs $0.90 \pm 0.05$, $p < 0.001$). The results are detailed in Table 1.

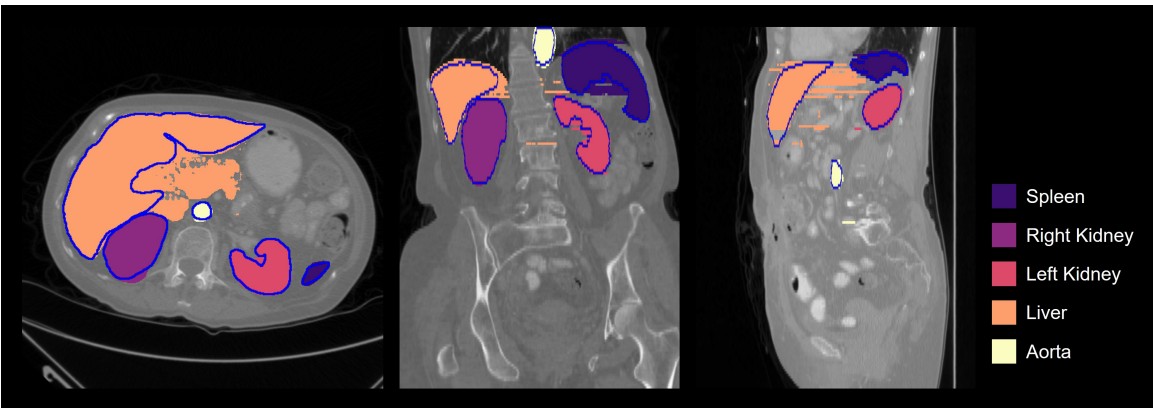

Figure 2: An example of crowd-sourced SAM-generated annotations for a CT scan from the BTCV train set in the axial, coronal, and sagittal views. The SAM-generated annotations are filled in while the ground-truth annotations are outlined in blue.

Table 1: Mean slice and volume Dice scores of SAM-generated annotations on the BTCV train set (left). Mean volume Dice scores of the GT-nnU-Net and SAM-nnU-Net on the BTCV test set (right). Best performing model highlighted in bold.

| Organ | Slice Dice | Volume Dice | GT-nnU-Net | SAM-nnU-Net | p-value |
|---|---|---|---|---|---|
| Mean | $0.88 \pm 0.02$ | $0.75 \pm 0.09$ | $\mathbf{0.90} \pm 0.05$ | $0.80 \pm 0.05$ | $< 0.001$ |
| Aorta | $0.88 \pm 0.01$ | $0.70 \pm 0.09$ | $\mathbf{0.92} \pm 0.01$ | $0.78 \pm 0.04$ | $< 0.001$ |
| Left Kidney | $0.88 \pm 0.04$ | $0.74 \pm 0.11$ | $\mathbf{0.87} \pm 0.12$ | $0.78 \pm 0.08$ | $0.02$ |
| Right Kidney | $0.90 \pm 0.02$ | $0.78 \pm 0.10$ | $\mathbf{0.87} \pm 0.11$ | $0.78 \pm 0.07$ | $0.06$ |
| Liver | $0.84 \pm 0.02$ | $0.73 \pm 0.13$ | $\mathbf{0.94} \pm 0.05$ | $0.84 \pm 0.04$ | $< 0.001$ |
| Spleen | $0.91 \pm 0.03$ | $0.80 \pm 0.14$ | $\mathbf{0.88} \pm 0.11$ | $0.80 \pm 0.11$ | $< 0.001$ |

## 4. Discussion

While SAM-generated annotations exhibit high mean slice Dice scores compared to ground-truth annotations, the lack of spacial relationships between features due SAM being designed for 2D segmentation results in sub-optimal performance for SAM-nnU-Net models. This can be addressed by developing adapting SAM for 3D segmentation to incorporate spacial relationships when generating segmentation masks (Lei et al., 2023; Wu et al., 2023; Bui et al., 2023; Gong et al., 2023). Another consideration is the potential for unreliable annotations from non-experts as seen with our missing annotations. Therefore, quality assessment is critical for filtering out low-quality annotations without manual intervention from an expert using metrics like inter-rater reliability or uncertainty estimation (Deng et al., 2023).

Limitations in current approaches warrant caution before incorporating crowd-sourced annotations. While we may not be ready for non-expert annotations yet, they have the potential for streamlining the annotation process for medical image segmentation by enabling anyone to annotate medical images from anywhere and at any time.

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
