# OpenReview forum: "Anytime, Anywhere, Anyone: Investigating the Feasibility of Segment Anything Model for Crowd-Sourcing Medical Image Annotations"
_MIDL.io/2024/Short_Papers — MIDL 2024 Short Papers_

### Official Review · Reviewer_CaA8 · 2024-04-23

**Confidence:** 5
**Final Rating:** 4

**Review:**

Summary
-------

This paper uses the Segment Anything Model (SAM) to generate training data (I.e. segmentation masks) from non-expert weak annotations. The authors then train an nnUnet on the SAM-generated training data and compare it to an nnUnet trained on ground truth segmentations. The model trained with ground truth labels substantially outperformed the SAM-nnUnet.


Strengths
---------

- The paper is very clearly written and the experiment setup and evaluation are sound.
- The authors have provided code to reproduce their experiments.


Weaknesses
----------

- Unfortunately, the proposed approach did not perform too well. However, even though this is a negative result, I believe this is an interesting and well-conducted study and can be of interest to the MIDL community.

---

### Decision · Program_Chairs · 2024-04-26

Accept